# Thermal Cycling of (RE)BCO-Based Superconducting Tapes Joined by Lead-Free Solders

**DOI:** 10.3390/ma14041052

**Published:** 2021-02-23

**Authors:** Michal Skarba, Marcela Pekarčíková, Lubomír Frolek, Eva Cuninková, Martin Necpal

**Affiliations:** 1Faculty of Materials Science and Technology in Trnava, Slovak University of Technology in Bratislava, Jána Bottu 2781/25, 917-24 Trnava, Slovakia; marcela.pekarcikova@stuba.sk (M.P.); eva.cuninkova@stuba.sk (E.C.); martin.necpal@stuba.sk (M.N.); 2Institute of Electrical Engineering, Slovak Academy of Sciences, Dúbravská cesta 9, 841-04 Bratislava, Slovakia; elekfro@savba.sk

**Keywords:** high-temperature superconductor, coated conductor, soldered joint, overlap joint, thermal cycling, lead-free solder

## Abstract

We prepared overlap soldered joints of high-temperature superconducting tapes, using various materials and preparation conditions. In order to select the joints with optimal performance, we correlated their electrical properties (derived from current–voltage curves) with the microstructure of the respective joint cross-section by scanning electron microscopy. With the first group of joints, we focused on the effect of used materials on joint resistivity and critical current, and we found that the dominant role was played by the quality of the internal interfaces of the superconducting tape. Initial joint resistivities ranged in the first group from 41 to 341 nΩ·cm^2^. The second group of joints underwent a series of thermal cyclings, upon which the initial resistivity range of 35–49 nΩ·cm^2^ broadened to 25–128 nΩ·cm^2^. After the total of 135 thermal cycles, three out of four joints showed no signs of significant degradation. Within the limit of 100 thermal cycles, reliable soldered joints can be thus prepared, with normalized resistivity not exceeding 1.4 and with normalized critical current above the value of 0.85.

## 1. Introduction

High-temperature superconductors with perovskite structure, for example (RE)Ba_2_Cu_3_O_7_ (RE = rare earth element), further reffered to as (RE)BCO, are often used in form of a thin layer. The functional superconducting layer has a ceramic nature, and thus for its use in thin tapes, it must be a part of a multi-layer sandwich composed mostly of metals and alloys.

While superconducting tapes (SCT) with lengths up to 1 km without a significant drop of critical current (*I_c_*) are routinely manufactured by major producers, even longer tapes are necessary for some devices. Therefore, a reliable soldering method for the fabrication of quality joints is required, as the properties of the joint can define the performance of the whole appliance [1]. Applications that need joints include superconducting fault current limiters (SFCL), transformers, coils, rotary machines, high-field magnets, and high-temperature superconducting (HTS) cables in general (e.g., cable-in-conduit conductor).

An ideal joint should have low overall joint resistance, low *I_c_* degradation, low manufacturing costs and time, reproducible preparation, adaptability for use in industry, good mechanical properties at both cryogenic and room temperatures, good adhesion, and good resilience against delamination or other damage upon mechanical and/or thermal cycling [2]. As there is no joint concept meeting all the requirements at the same time, multiple solutions are currently developed in parallel. Examples of concepts are variations of (1) different geometries: overlapped (Table 1), butt [3], and bridged [4]; (2) different joining techniques: soldered joint (Table 1), ultrasonic welded joint ([2,5]), inserted mechanical lap joint [5,6], diffusion joints [7], and (3) different resistivity: resistive or superconducting [8,9].

In field of superconductors, soldering has been successfully used to connect SCT with (1) conventional conductors, (2) with a stabilizing shunt layer, (3) with the other end of the same tape to form a loop ([18,26]) and (4) with other SCT. In this study, we examine the latter joint type. Despite its obvious drawbacks, the production of soldered joints remains one of the most used methods for industrial applications, thanks to their relatively simple and quick fabrication, and reasonable cost. The main disadvantage of soldered joints is their relatively high resistivity, even if only compared with other resistive (non-superconducting) joints.

The overall resistivity of a soldered joint between two SCTs ([25] and [27,28,29,30,31], see Figure 1) consists of the following contributions [16]: (1) resistivities of stabilizing layers (Cu, Ag), (2) resistivity of the solder interlayer itself, and (3) interface resistivities between the (RE)BCO/Ag, Ag/Cu, and Cu/solder layers. While the resistivities of components (1) and (2) are estimated in the order of (100^−1^–10^0^) nΩ·cm^2^, the interface resistivities (3) dominate the overall solder joint resistivity at the order of 10^1^ nΩ·cm^2^ [12,21]. In case the joint resistivity is inappropriately high, the joule heating may significantly reduce the *I_c_* at the joint region and increase the consumption of coolant [21].

For the application of soldered joints in some devices, such as SFCL, one of the most important properties of joints is their performance (temporal evolution of joint resistivity and critical current) after a certain amount of quenching events during fault current limitation. In order to simulate the quenching in liquid nitrogen (LN_2_) environment (at its boiling temperature at atmospheric pressure), we performed series of thermal cyclings of the soldered joints of SCT to gain more insight into the correlation of the joint electrical properties with the number of the LN_2_ → 150 °C thermal cycles performed with the joints.

There are numerous studies on the evolution of the microstructure of a low-melting temperature solder itself (e.g., Sn96.5Ag3Cu0.5) upon cooling, thermal cycling, or aging, e.g., [32,33,34,35]. Studies aimed at the preparation of a good quality soldered joints of SCTs are reviewed in the next paragraph, with characterizations of what we further refer to as an “initial state”, which is a state before thermal cycling. Nevertheless, to our best knowledge, there are only a few publications extended also with the thermal cycling of soldered joints of SCT; we discuss them at the end of the following literature overview.

Baldan et al. [1] showed that the increase of the overlap length of a joint (in the range of 1 to 20 cm) caused a monotonic decrease of *R_j._A_j_*. They experimented with different joint geometries and surface preparation processes, with and without application of pressure during the soldering process. In their follow-up study [14], they also investigated the mechanical properties of joints (tensile stress); similar measurements were published by [11], in which they found that etching out the SCT Cu stabilizer reduced both the *R_j._A_j_* and *I_c_* degradation upon axial load. Shin et al. [10] investigated the degradation of joint’s *I_c_* after uniaxial tension. From all the relevant literature, we found that they achieved the lowest value (among joints with overlapped length of few cm) of *R_j._A_j_* = 3.6 nΩ·cm^2^. Different soldering methods and temperatures at which current*–*voltage (*I–V)* curves were measured were investigated by [21]. They found a strong dependence of *R_j._A_j_* on the SCT batch. Lécrevisse et al. [13] investigated the dependence of *R_j._A_j_* on the batch and joint overlap length; they also present a method to minimize the degrading effects of solder heating on *I_c_* and the dependence of the resistance of the solder materials. Kim et al. [17] examined the effects of pressure, joint length, solder composition, and SCT batch on the joint resistivity. They found that SCT batch and solder were the primary sources of a wide range of variation in the joint resistivity; on the other hand, there was no significant difference when comparing three 4 mm wide SCT cuts from one 12 mm original tape. Tsui et al. [20] found that the overall joint resistivity predominantly comes from the solder layer and the interfacial resistivity at the Ag–(RE)BCO interface. A systematic study of the joints, in which they the varied temperature, magnetic field, and solders was coupled with numerical modeling of voltage and temperature profiles. Bagrets et al. [16] correlated the overall resistances of (RE)BCO-based SCT lead-soldered overlap joints with interface resistance between the layers. The interfacial resistance, which was determined by evaluation of the current transfer length measurement, showed to be the main contributor to the overall resistance of the soldered joint. Balashov et al. [12] found an inverse proportionality between the pressure used for the preparation of joints and the joint resistivity. Pressures up to 8 MPa had no degradation effect on *I_c_* and, depending on the tape manufacturer, they managed to prepare the joints with resistivity down to 10 nΩ·cm^2^. Han et al. [18] stressed the importance of the flatness and smoothness of the soldering table for the preparation of low resistance joints. They prepared a series of joints with different solder layer thickness and overlap lengths, and they confirmed that the decrease of solder layer thickness is correlated with the reduction of *R_j._A_j_*. Drienovsky et al. [22] showed a difference in microstructure between reflow soldered joints and induction soldered joints. In the latter approach, the heat is localized only in the solder itself and acts for a shorter time compared to conventional ovens (<20 s). Compared to traditional heating, the resulting structure of an induction solder layer contains less voids and coarse Cu_6_Sn_5_ particles. Zhang et al. [15] systematically investigated the effects of soldering parameters (temperature, pressure, surface treatment) and solder materials on the electrical and mechanical properties of the joints. The overlap length was as long as 300 cm.

In our previous study [24] on overlapped joints based on lead-free solders, we constructed a clamping device (Appendix A) to have a control over the soldering pressure and the evolution of temperature. The work was focused on the microstructure of the soldered joint layer as a function of used flux, pressure, and reflow time. For one joint, the thermal cycling was also performed.

To our best knowledge, there are only two more studies that include thermal cycling (or aging) of joints similar to those prepared in this work. Balashov et al. [23] investigated low-resistance overlap joints of two types of SCT, using two types of solders. The joints were subjected to 10 thermal cycles from LN_2_ temperature to 80 °C, causing fluctuations of *R_j._A_j_* only within a few percentage points. Then, the solder was re-melted, and using pressure, the thickness of the soldered layer was decreased, which significantly lowered the *R_j._A_j_*. Preuss et al. [26] investigated *I_c_* degradation at soldering conditions by thermal aging due to oxygen diffusion from the (RE)BCO layer, which was simulated by heating the joints in a solder bath for 30 s up to 2 h. While the *I_c_* decrease after 10 min of heating was about 5% at 225 °C, it increased to 45% at 300 °C. The degradation was monotonically increasing with time and temperature, but it was in general independent of the number of thermal cycles (TC).

In our present work, we studied two groups of face-to-face overlapped non-superconducting soldered joints of two pieces of second generation coated superconductor, with varying preparation conditions (clamp pressure, quenching temperature), materials (SCT, solder, flux, and cleaning agent) and number of performed thermal cycles.

## 2. Materials and Methods

### 2.1. Materials

#### 2.1.1. Superconducting Tapes

For the fabrication of all joints in this study, we used a commercially available high-temperature SCT. The initial critical currents (*I_c0_*) of all joints did not decrease very significantly below the minimum guaranteed value reported by the producer for the whole length of supplied spool (*I_cprod_*). The latter fact indicated a good quality of the superconducting layer itself.

Both types of SuperPower 2G HTS tapes were based on (RE)BCO, had a Hastelloy substrate, and were purchased from Furukawa Electric Europe Ltd., London, United Kingdom. Tapes were fully coated with copper finish as a surround stabilizer. The type SCS4050-AP we further denote as “S4”, while the SCS6050-AP is referred to as “S6”. Values of *I_cprod_* and the *n*-value given by the producer are listed in Table 2. As some values of layer thicknesses considerably differed from those declared by the supplier, in Table 2, we only report layer thicknesses measured by us from SCT cross-sections by scanning electron microscope (SEM).

#### 2.1.2. Solders

One of the most important parameters for the selection of appropriate solders was their liquidus temperature, which should be (1) not too low as not to melt the joint during a quench event, but (2) on the other hand, the liquidus should not be too high in order not to degrade the superconducting properties of (RE)BCO during the joint fabrication. The highest temperature that the SCTs experienced in this study during reflow was 245 °C, which was only slightly above the maximum value of 240 °C recommended by the SuperPower company [37]. According to [38], 0.6 μm thick YBCO (which is (RE)BCO with (RE) = yttrium) coated film heated for 30 min at 250 °C undergoes 10% increase of its resistance at room temperature due to oxygen diffusion out of the epitaxial film, while the change of critical current density was negligible. Therefore, we expect that the heating of SCTs (from any manufacturer) to 245 °C for tens of seconds should not cause significant (RE)BCO degradation.

Regarding the solder shape, [17] found that a ribbon solder results in the most reproducible joint resistance; [21] concluded that ribbon solders gave at almost all conditions lower *R_j._A_j_* than pre-tinned joints. Therefore, both solders used for production of the joints between two SCTs were ribbon-shaped with thickness of 55 μm. Particular properties for each used solder are summarized in Table 3.

#### 2.1.3. Fluxes and their Deposition

Fluxes listed in Table 4 were selected according to the results of preliminary peel tests [40]. In all cases, the fluxes were deposited at the surfaces of both jointed tapes.

#### 2.1.4. Cleaning Agents

Acetone per analysis purity(p.a.) was used as a degreasing agent for the copper surface of SCT before soldering of the joints G1.IN4 and G1.SN3 (see joint labels in Table 5). For all other joints, to degrease and remove copper oxides, we used Nital, which was 2 vol% HNO_3_ dissolved in 99.8 vol% ethanol.

### 2.2. Methods

#### 2.2.1. Preparation of the Joints

All joints in this study were prepared by soldering two pieces of SCTs under applied pressure, with the materials and soldering conditions listed in Table 2, Table 3, Table 4 and Table 5, according to the following procedure. Two 8 cm long parts of copper-finished SCT were cut from the same tape batch (typically neighboring pieces) by surgical scissors, and the surface areas not intended for soldering were covered with a Kapton tape, to prevent oxidation (during reflow) and contamination with excess flux and solder. Surface parts intended for overlapping were cleaned with Nital or acetone and then immediately covered with a thin layer of flux. Rectangles with dimensions of 30 mm × 3 mm or 30 mm × 5 mm were cut from the solder ribbon for soldering of 4 mm or 6 mm wide SCT, respectively. Then, the joint was assembled in a face-to-face way, as sketched in Figure 1, and installed into the bottom of a home-made Al-alloy clamping device (Appendix A). Then, clamp pressure was set by varying the length of calibrated springs, exerting force onto the joint via a metallic block. The base dimensions of the metallic block were 3 × 30 mm^2^ or 5 × 30 mm^2^ for 4 mm or 6 mm wide tapes, respectively. The 0.5 mm rim at each side of the assembly width served as a space for the leakage of excessive solder. The block was in direct contact with the upper surface of the joint assembly, evenly distributing the pressure from the clamp springs and holding a plug-in thermocouple. Then, the whole clamp with the assembly of tapes was put onto a hot plate, heated up to 400 °C, and the recording of the block temperature was initiated. The soldering was terminated as soon as the quenching temperature (i.e., the maximum temperature reached during soldering), *T_q_*, was reached. The quenching was performed by immersing the clamp to a water bath with room temperature, in order to minimize the growth of large, primary solidified intermetallics [34]. Reflow time, *t_l__→q_*, was recorded, which was a time between reaching the liquidus temperature (*T_liquidus_*) and quenching. Then, the joint was dismounted, dried, cleaned with alcohol, and mounted to a fiberglass–plastic composite holder by soldering it to two current and two voltage terminals, using the “FL” solder.

Finally, we verified the quality of a new joint by measurement of its critical current. The latter yielded values of *I_c0_* for each joint, which was the critical current before the thermal cycling (that is, at 0 TC). We further used the values of *I_c0_* as a reference to calculate the normalized *I_c_*/*I_c0_* values for the thermally cycled joints. We accepted a newly prepared joint only in case its *I_c0_* was at least 80% of *I_cprod_*.

Thus, soldered joints were prepared, each 13 cm long, with 3 cm of face-to-face overlap in the middle [12], with a joint area of 120 mm^2^ or 180 mm^2^ for 4 mm or 6 mm wide tapes, respectively (Appendix A). In the whole course of joint preparation, attention was always paid to (1) minimize the time the (RE)BCO layer experiences temperatures above 200 °C, which is the onset of the (RE)BCO degradation ([17,38]) and (2) the joints would experience minimum mechanical load, such as bending or tension.

#### 2.2.2. Cross-Sections for SEM Analysis

Parallel with the joints intended for *I–V* measurements, we used the same method and materials to manufacture a joint copy, which only served for the preparation of a cross-section. All cross-sections samples prepared from the Group 1 joints (that is from G1.IN1 to G1.S6 in Table 5) were cut from the plane perpendicular to the current flow by a laser ablation machine LASERTEC 80 Shape (DMG MORI, Tokyo, Japan), which was followed by polishing with Ar ions using the cross-section polisher device JEOL SM-09010 (JEOL, Tokyo, Japan). During the performance of thermal cyclings, the respective cross-section was always thermally cycled along with the corresponding joint intended for *I–V* measurements. SEM analysis was performed with a JEOL JSM 7600F (JEOL, Tokyo, Japan) setup, which was equipped with an energy-dispersive X-ray (EDX) detector for chemical analysis.

SEM micrographs were used for analysis of the joint structure, including its porosity. Porosity was roughly estimated as a ratio between the sum of areas identified as voids to the total area of the solder layer in the SEM micrographs.

#### 2.2.3. Thermal Cycling

The type of thermal cycling we used was a compromise between two requirements: (1) to ensure conditions close to the ones expected for the most probable applications of the joints, that is in an SFCL, and (2) to use a method potentially usable in industry, with stress to the time requirements and costs.

A thermal cycle begun with heating of the joints to 150 °C in a drying oven in air atmosphere. Then, the joints were immersed into LN_2_, until the violent boiling of LN_2_ at the surfaces vanished. Afterwards, the joints were shortly kept at room temperature for visual checking of any mechanical damage at the surface of joints. Thus, the joints experienced a temperature difference of 346 °C each cycle. After a predetermined number of cycles, the joints proceeded to the *I–V* curve measurement. Should the *I–V* measurement show no dramatic degradation of *R_j._A_j_* or *I_c_*, another set of thermal cyclings (TC) was performed.

In addition to the aforementioned TC from LN_2_ to 150 °C, due to experimental reasons, the joints also experienced two other types of thermal cyclings (which we will refer to as “minor TC“) with temperature difference < 250 °C, namely: (1) minor TC influencing the whole joint, namely the *I–V* measurements, during which the joints had to be fully immersed in LN_2_ and (2) potential thermal shocks during soldering, having local thermal effects: (a) the manufacture of a soldered joint between the tapes, and (b) soldering of joint terminals onto the terminals of holders (Figure 1). While the Group 1 joints were soldered to holders before each *I–V* measurement, and then de-soldered from all four terminals before the next set of TC, we performed the TC of the Group 2 joints (that is joints from G2.SD1 to G2.SL2 in Table 5) without dismounting the joints from the holders. Thus, the reduction of soldering steps could contribute to a decrease of thermal and mechanical degradation. On the other hand, TC with holders (having much higher heat capacity than the joint itself) slowed down the kinetics of heat exchange and thus increased the dwell times at each stage of TC, as listed in Table 6.

##### 2.2.4. I–V Measurements

To record the current–voltage curve, we used the four-probe method routinely used in similar studies [23]. All *I–V* measurements were performed in an LN_2_ bath (−196 °C), in self-field, with the threshold electric field criterion of 1 μV/cm used to determine the critical current. A more detailed description of the method and analysis of the results is provided in our previous study [24].

As a criterion for a good quality joint, we chose *R_j._A_j_* ≤ 50 nΩ·cm^2^ and *I_c_* ≥ 0.8 *I_c0_*. Regarding the resistivity limit, we based the value on previous studies (for similar materials and conditions): Balashov et al. [23] considers low resistivity for joints at 77 K in range of (30−50) nΩ·cm^2^; [41] states that a typical solder joint resistivity for SuperPower SCT is (40−50) nΩ·cm^2^ for a Cu stabilized conductor measured at 77 K in self field; according to [13], a resistivity range of (30–60) nΩ·cm^2^ is common for the SuperPower joints (using PbSn solders), while for the SuNAM joints (SuNAM Co*.,*Ltd*,* Gyeonggi, Korea), a typical value is about 100 nΩ·cm^2^ (see also Table 1).

## 3. Results and Discussion

In the search for optimal materials and conditions for the preparation of a good quality soldered joint, we investigated 15 joints in total, which were divided into two groups, according to the particular objective investigated. With the Group 1 testing joints, our main goal was to select the materials and optimize the preparation procedure in order to prepare a joint with acceptable initial *R_j._A_j_* and *I_c_*. With Group 2, which only contained good quality joints, we performed a series of thorough thermal cyclings until the first signs of serious degradation of *R_j._A_j_* and/or *I_c_* appeared.

### 3.1. Group 1 Joints: Selection of Materials and Optimization of the Preparation Procedure

#### 3.1.1. Initial State of the Joints

Within this group, 11 joints were prepared, differing in solder, flux, cleaning agent, and clamp pressure, in order to obtain a joint with *R_j._A_j_* ≤ 50 nΩ·cm^2^ and *I_c_* ≥ 0.8 *I_c0_* at its initial state (that is before thermal cycling). The materials, soldering conditions, and results of the initial *I–V* measurements are summarized in Table 7.

##### 3.1.1.1. Joints with the “IN” Solder

For the first joint (G1.IN1), we chose the indium-based solder “IN“, as it was the solder with the lowest resistivity available (compared mainly with SAC solders), and moreover, the ”IN“ solder had the lowest available liquidus temperature, so that the (RE)BCO layer would experience the lowest deterioration from the effect of elevated temperature during soldering. Although there are studies with several sorts of In-based solder used for copper surface, e.g., [19], disadvantage of In-based solders is their worse mechanical properties. In-based solders are not recommended for the soldering of copper in general, as In and Cu diffuse into one another, forming a brittle intermetallic. For the joint G1.IN2, exactly the same preparation procedure was copied as for the G1.IN1, in order to also test the reproducibility of our experimental preparation method. The relative change of initial *I_c0_* for both G1.IN1 and G1.IN2, with regard to the respective *I_cprod_* was acceptable (−6% and −11%, respectively, see Table 7). Deviations of electrical properties (*I_c0_* and *R_j0._A_j_*), when compared between the G1.IN1 and G1.IN2 joints, showed a very good reproducibility. However, the absolute values of *R_j0._A_j_* above 175 nΩ·cm^2^ were out of the desired limit. We suspected that the macroscopically observed evolution of a vapor from the flux during the joint fabrication could be the main reason for the high value of *R_j0._A_j_*, as the vapor could cause the porosity of the solder layer. Indeed, significant pores were found in SEM micrographs in the joints G1.IN1 and G1.IN2 (not shown; will be discussed in the next paragraph).

In an attempt to reduce the *R_j._A_j_*, we used another soldering flux (the joint G1.IN3) and increased the quenching temperature to 180 °C, aiming to increase the solderability. The “SD“ flux with a consistence of a thick gel offered not only more reproducible deposition, but unlike with the previously used flux “F2“, no gas emerging from the joint during reflow was noticed. Nevertheless, as evident from Figure 2, very similar pores in the solder layer are also typical for the joint G1.IN3, in which they span almost across the whole length of the cross-section.

In the joint G1.IN4, we investigated another two modifications possibly leading to a decrease of the high *R_j._A_j_*: (1) We increased the soldering clamp pressure, which may repeal the gas bubbles (if present) in the molten solder more effectively and possibly decrease the soldered layer thickness, as was also suggested by [12,18]; and (2) we modified the method for cleaning of the tape copper surface, expecting that a more thorough removal of corrosion products could decrease the contact resistivity. Although it has been confirmed by others that the pressure reduces and homogenizes the joint thickness [1], neither of the aforementioned modifications had any significant effect on *I_c0_*, and the value of *R_j0._A_j_* for the joint G1.IN4 of 181 nΩ·cm^2^ suggests that no resistivity decrease was observed, compared to the joints prepared so far, which ranged from 162 to 178 nΩ·cm^2^.

##### 3.1.1.2. Joints with the “SN” Solder

In preparation of the joints so far, we have varied the flux, cleaning agent, and the clamp pressure, but we used only one type of solder. For preparation of all the following joints of this group, we changed the used solder to the “SN“, that is, an SAC-type (composed of Sn, Ag, and Cu) solder. In comparison with the “IN“ solder, the “SN“ solder had significantly higher liquidus temperature and almost double the resistivity at room temperature (Table 3); both features could potentially have a negative effect on the *R_j0._A_j_* of a joint. On the other hand, the “SN“ solder is supposed to be more appropriate for the soldering of copper; thus, less structural features in the solder layer increasing the *R_j._A_j_* could be expected. In the following subgroup of the Group 1 joints, we focus mainly on varying the type of flux.

The parameters of the joint G1.SN1 were almost the same as for the G1.IN1, except for the solder. Although a typical microstructure of the G1.SN1 joint cross-section (Figure 3a) shows less significant voids compared to the “IN“ joints (Figure 2), compared to an average of 175 nΩ·cm^2^ for the four “IN“ joints, the *R_j0._A_j_* has almost doubled to 323 nΩ·cm^2^ for G1.SN1. Unlike in case of the “IN“ joints, areas of the cross-section could be easily found with almost no voids in the solder layer of the joint G1.SN1. The right-hand side of micrograph Figure 3a shows the significant porosity and non-uniformity of the Ag layer thickness, which varied in range from approximately 0.1 to 2 μm.

In the joint G1.SN2, which was a repetition of the joint G1.SN1 preparation with “R5“ flux instead of the “F2“, the *R_j0._A_j_* decreased to 220 nΩ·cm^2^. In this joint, the highest porosity in the joint was observed (not shown).

The joint G1.SN3 had similar value of *R_j0._A_j_* with the previous one. The joint G1.SN3 was a repetition of the joint G1.SN1 with the use of “F2“ flux, but (1) using acetone instead of Nital as cleaning agent, and (2) using the highest value of clamp pressure in this study. As it will be discussed further, the modified parameters had an effect neither on the decrease of soldered layer porosity nor its thickness.

Although the joint G1.SN4 was prepared in a very similar way as the G1.SN5 (except the reflow time, cf. Table 5), the variance of their *R_j0._A_j_* spanned almost the whole range for all the “SN“ joints of this group (the joint G1.S6 is excluded from this comparison), regardless of the used flux or other preparation conditions, which points to the unsatisfactory reproducibility of joints preparation. The main point of the G1.SN4 and G1.SN5 joints was to reduce the porosity of the solder layer as much as possible. To this end, we (1) used solid rosin with no added solvent as flux, to avoid potential void forming by vaporizing of the flux solvent during reflow, and (2) we performed the reflow in two steps: a clamp pressure of 175 kPa was set at the first step; after the reflow temperature was reached (Table 5), the joint was quenched, and the pressure was increased before the second reflow to 309 kPa, in order to force as many bubbles as possible out of the solder layer. Two features should be noted from the cross-section of the joint G1.SN5 (Figure 3b): (1) the prevalence of the Cu_6_Sn_5_ phase due to the relatively low thickness of the solder layer (<4 μm). The lack of the β-(Sn) phase could have an effect on the mechanical properties of the joint [42]. (2) While the porosity of the solder layer in all joints prepared so far was above 75% (see Table 7), for the joints G1.SN4 and G1.SN5, it decreased dramatically to about 20 for the first time.

The latter feature might support our hypothesis that the main factor responsible for the excessive formation of voids during reflow could be the volatile ingredients of the flux. The pressure difference between the two reflow soldering steps was probably too low to reduce significantly the concentration of voids, as the study [43] suggests that an effective ratio between the two pressures would have to be in the order of at least 100. One of the best contacts of a solder with jointed surfaces was found for the joint G1.SN6, as shown in Figure 3c. The solder layer is one of the thinnest of all investigated joints, has minimum voids, and consists prevalently of the Cu_6_Sn_5_ phase. However, no significant improvement in the *R_j._A_j_* was achieved.

The best value of *R_j0._A_j_* for all 10 joints discussed so far was still more than three times higher than the desired value of ≤ 50 nΩ·cm^2^. As a result of the occurrence of the above-mentioned structural defects at internal interfaces of the SCT (which we suppose were inherent to the tape and not influenced by the soldering process), we repeated the preparation of the last joint (G1.SN6) with a similar SCT, namely the “S6“ (the most significant difference was another tape batch with different width). Thus, joint G1.S6 was produced (Figure 3d), with an *R_j0._A_j_* value of 41 nΩ·cm^2^, which is seven times lower than the average of the joints G1.SN1 to G1.SN6, despite the fact that the porosity of the solder layer was higher than that of joint G1.SN6. If we also take into account considerably the lower roughness of the Ag layer (the one covering (RE)BCO) in the joint G1.SN6 (compared to all joints made of the S4 tape), we might draw a suspicion that the high resistivities of “S6” joints come from interface resistivities within the tape itself rather than from defects in the solder layer.

##### 3.1.1.3. Group 1 Joints: Summary of the Initial State

To summarize the properties of the Group 1 joints in their initial state (before thermal cycling), their quality (in terms of their *R_j0._A_j_*) was most strongly influenced by the quality of the SCT itself as received from the producer, but it was only moderately dependent on the structure of the solder layer. The strong dependence of *R_j._A_j_* on batch was reported by multiple previous studies e.g., [21], or [17]. In the latter study, the authors achieved *R_j._A_j_* as low as 28 nΩ·cm^2^ for some batches, while for other batches, it was impossible to achieve a joint resistivity < 100 nΩ·cm^2^. As cutting a long tape to batches by the manufacturer can be arbitrary, we suppose that a similar fluctuation of interfacial resistivities could naturally occur also within one batch, as it was experienced in our case—except for the outstanding joint G1.S6, the resistivities ranged from 163 to 341 nΩ·cm^2^.

Our values of *R_j0._A_j_* were much more sensitive to the occasional occurrence of local defects at the (RE)BCO–Ag, Ag–Cu, and buffers–Hastelloy interfaces than to excessive porosity of the solder layer. Due to the overwhelming effect of the internal tape resistivities, we observed almost no correlation between the *R_j0._A_j_* and porosity of the soldered layer (Appendix A), and no clear trend could be observed between the *R_j0._A_j_* and the thickness of the soldered layer (Appendix A).

Complementary to the values of *I_c_*, the *n-*values also reflect the performance of the (RE)BCO layer itself. The *n-*value is based on a statistical distribution of critical currents in the superconductor and is thus directly proportional to the macrostructural and microstructural homogeneity of (RE)BCO [36]. In this regard, a whole study only dedicated to the correlation of the (RE)BCO structure with *I*−*V* measurements would be useful, as simple considerations (e.g., the effect of soldering temperature) without thorough structural analysis fail to explain why the median value of *n*-values for the “IN” soldered joints are considerably lower than the median for the “SN” soldered joints (23 vs. 29, respectively).

The solder layer was typically rich in voids, but while the “SN“ solder areas with no defects were easy to find, such places were not found at all when the “IN“ solder was used. For the use of the “IN“ solder with copper finished SCTs, we therefore conclude that the solder would be probably more appropriate for Ag-finished tapes despite the systematically lower *R_j0._A_j_* values compared to the “SN“ solder joints.

Previous studies pointed out that some level of porosity at the Cu/”SN” solder interface cannot be avoided due to the coalescence of atomic-level vacancies into the so-called Kirkendall voids [44,45,46,47]. According to [47], the Kirkendall voids could be formed due to electromigration effects, where high current densities (in order of 10^8^ A/m^2^) are present. Moreover, [43] determined the experimental lower limit (the “residual porosity”) for void content in an SAC solder to be 4.8 vol%. Obviously, the performance (such as the resistivity and lifetime) of the solder joint with large or many voids could be considerably reduced, but the reduction also depends on the frequency and location of voids [48]. A way to decrease the porosity might be to increase the grain size in the Cu overlayer [49].

The structure of the solder layer for all Group 1 joints was moderately correlated with the chemical composition of the flux (gas bubbles from flux solvent evaporation), but it was related neither to the used cleaning agent nor to the set clamp pressure. Regarding the latter parameter, in the dependence of the thickness of soldered layer as a function of clamp pressure (Appendix A), we observed no correlation, which could be explained by the fact that both the range and the absolute value of pressures achievable with our clamp were (compared to other studies) rather low. There are studies in which authors try to avoid increased pressure during soldering in order to achieve a better reproducibility of results [20]. In the latter case, the increase of pressure as a means to achieve thinner solder layers with lower joint resistivities was refused. On the other hand, other authors reported the successful use of soldering pressure of 6−7 MPa [16], 15 MPa [23], and even up to 100 MPa [6]. In any case, despite our observation of the low effect of porosity on the electrical properties of a joint, it might be useful to keep the porosity at a reasonable level when mechanical properties come into consideration, especially after the thermal cycling of a joint.

The plot of porosity of the soldered layer as a function of clamp pressure showed no correlation (Appendix A). Thus, we rather tried to reduce the porosity of the soldered layer not by quantitative changes of pressure, but rather qualitative: by two-step pressure ramping (the joints G1.SN4 and G1.SN5).

Concerning the different reflow times used for the preparation of the Group 1 joints, they seemed to be uncorrelated with the thickness of the soldered layer.

As mentioned in the “Methods“ section, cross-sections samples for SEM imaging were not made (due to the destructive nature of sample preparation) of physically the same joint as was used for the *I–V* measurements. Assuming the potentially changing properties of SCT along its length, the fact that the cross-section is only a replication of the “*I–V* measured joint“ could decrease the correlation between the joint structure and its electrical properties. Therefore, in our future studies, a non-destructive tomography method would be probably a more appropriate alternative to the SEM analysis.

#### 3.1.2. Thermal Cycling of the Joints

Except for the joint G1.S6, all other Group 1 joints were subjected to 75 TC or 100 TC, for the “IN“ solder joints or the “SN“ solder joints, respectively.

##### 3.1.2.1. Joints with the “IN” Solder

Figure 4 shows the evolution of electrical properties of the “IN“ solder joints as a function of the number of thermal cyclings. For easier comparison, all properties on the vertical axes were normalized with their respective value at the initial state (at 0 TC). After 25 TC, the “IN“ joints showed none or only a slight degradation of electrical properties, but after 50 TC, the normalized resistivity (Figure 4a) of the G1.IN1 joint significantly increased. As a reason for the increase of *R_j._A_j_*, we would expect thickening of the Cu_11_In_9_ intermetallic layer with the increasing number of TC. Although in total, three out of four of the “IN“ joints exhibited an increase of *R_j._A_j_* with thermal cycling, SEM micrographs showed no Cu_11_In_9_ layer thickness increase.

The (RE)BCO layer of G1.IN4 seemed to be seriously damaged after 50 TC, as suggested by drop of its normalized critical current (Figure 4b), but then, surprisingly, at 75 TC, the latter property recovered close to its initial value. After 75 TC, based on the *I_c_*/*I_c0_* reduction, we must conclude that two out of four joints exhibited major degradation. Even though the selected areas of the soldered layer were carefully compared before and after thermal cycling (not shown), no changes in the structure (especially the porosity) of the soldered layer of the “IN“ joints could be observed by SEM. We conclude that the thermal cycling caused no significant structural changes to the solder layer observable by SEM imaging and chemical analysis.

We thus suspect that it was rather the growth of defects at the interfaces of SCT layers that could be the main cause of *R_j._A_j_* and *I_c_* degradation, although more microstructural evidence would be required to corroborate this hypothesis.

##### 3.1.2.2. Joints with the “SN” Solder

Six joints soldered with the “SN“ solder were subjected to 100 TC; the remaining joint (G1.S6) had outstanding properties and will be discussed later. In the course of thermal cycling, we expected to observe some changes in *R_j._A_j_*, as a result of (1) the evolution of defects present already in the initial state, and/or the formation of new defects, in all layers of SCT, and (2) structural changes in the solder layer, mainly the growth of intermetallics with various resistivities (Figure 1). To evaluate the two types of structural changes, 36 different areas were analyzed by SEM and EDX in total for the six joints, with the step of 25 thermal cycles.

Regarding the first point, which had defects in all layers of SCT, the type of defects were changing in the course of thermal cycling, dominantly at the lower number of TC: (a) formation and growth of voids within the solder layer, with diameter of few μm, e.g., as in Figure 5; (b) and at the Ag/Cu SCT interface, e.g., as in Figure 6.

Concerning the structural changes in the solder layer, we supposed that the growth of intermetallics would lead (at early stages of thermal cycling) to the increase of *R_j._A_j_* as a result of the Cu_6_Sn_5_ layer thickening, and at a higher number of TC, the *R_j._A_j_* would eventually decrease due to the thickening of the Cu_3_Sn phase, at the expense of the Cu_6_Sn_5_ growth. The coagulation and enlargement of Ag_3_Sn particles could also take place. However, even after 100 TC, we observed only slight thickening of both intermetallic layers (Figure 7), but the Ag_3_Sn particles expanded in volume at the expense of their amount (Figure 5, Figure 6 and Figure 7). In addition, another phase transition effect (unrelated to intermetallics) was rather apparent predominantly in the thicker joints (Figure 5): the swelling of the solid solution (Sn) phase, which was visible at its interface with the Cu_6_Sn_5_ intermetallic and was most probably due to the onset of allotropic transformation of β-form metallic (Sn) to the non-metallic (and less dense) α-(Sn). In case all the β-(Sn) would transform to α-(Sn), its volume would increase by 27% [50]; therefore, we suppose that we only observed the early stages of the phase transition. The swelling was clearly an effect of thermal cycling, as we never observed it in the initial state. The β-(Sn) to the α-(Sn) transformation caused cracks in the Cu_6_Sn_5_ and Cu_3_Sn phases only in one out of eight investigated cases, namely in the G1.SN1 joint. Naturally, the β-(Sn) to the α-(Sn) transformation only concerned the joints with a thicker solder layer, where a sufficient volume ratio was occupied by the β-(Sn) phase. The so-called “tin pest” phenomenon deserves more attention, as a degradation of mechanical properties in lead-free soldered joints was also confirmed e.g., in [51].

Evolution of the electrical properties of the “SN“ solder joints, as a function of the number of thermal cyclings, is shown in Figure 8. *R_j_*/*R_j0_* is monotonously increasing for the majority of the joints (Figure 8a). The increase was probably an outcome of the defects displayed in Figure 5, Figure 6 and Figure 7. After 100 TC, five out of six joints still had value of *R_j_*/*R_j0_* in the satisfactory range of 1.02 to 1.48. The joint G1.SN2 was seriously damaged after >75 thermal cycles, as also confirmed by its *I_c_*/*I_c0_* (Figure 8b) dropping to 0.02, and *n/n_0_* decreasing to 0.13 (Figure 8c). According to the value of *I_c_*/*I_c0_*, after 100 TC, only two out of the six joints preserved a value > 0.8 (our criterion for a serious joint degradation), namely the joints G1.SN5 and G1.SN6 (until < 50 TC, all six joints would meet the > 0.8 criterion).

Compared with the “IN” soldered joints, the *n-*values were significantly more scattered for the “SN” joints (cf. Figure 4c and Figure 8c), with increasing trend toward the higher amount of TC.

##### 3.1.2.3. Group 1 Joints: Summary of Thermal Cyclings

In case joints with not very high performance should be prepared, that is (1) resistivities up to 341 nΩ·cm^2^ are sufficient, and (2) reliable operation only up to 25 TC is required, the proposed materials and methods are appropriate, as they are low cost and relatively quick. Resistivities down to 163 nΩ·cm^2^ could be achieved in case of the “IN“ solder, with the range of reliable operation limited to 25 TC; should a higher amount of TC (and perhaps better mechanical properties) be necessary, the “SN“ solder should be preferred, even though they possessed the drawback of doubled resistivity compared to the “IN“ joints. However, as suggested by the properties of the joint G1.S6, it seems that merely by paying attention to the quality of internal SCT interfaces before soldering, it is possible to produce a joint with resistivity < 50 nΩ·cm^2^. As we will show in the following section, a series of joints could be prepared in the latter way, with joint resistivities consistently below 50 nΩ·cm^2^, as well as with higher resilience against thermal cycling.

### 3.2. Group 2 joints: Performance of Good Quality Joints during Thermal Cycling

#### 3.2.1. Initial State of the Joints

In the previous section, we characterized the soldered joints with the *R_j._A_j_* in the order of 10^2^ nΩ·cm^2^. With the Group 2 joints, it is now our goal to show that both the resistivity and resilience of the joints against thermal cycling can be substantially improved, provided that SCT with low resistivity between internal tape interfaces is used.

The design of all four Group 2 joints was based on the joint G1.S6, which had the *R_j._A_j_* = 40.6 nΩ·cm^2^. Materials, soldering conditions, and initial *I–V* measurements are summarized in Table 5 and Table 8, respectively. We prepared the joint G2.SD1 using the “SD” flux; the joint G2.SD2 was a replication of G2.SD1. Flux “SL” was used for the joint G2.SL1; the joint G2.SL2 was a replication of the joint G2.SL1. Thus, apart from the flux, all the Group 2 joints were prepared in the same way in order to also verify the reproducibility of the joint properties.

The Group 2 joints had *R_j0._A_j_* in the range from 35.3 to 49.4 nΩ·cm^2^ (Table 8). Except for the joint G2.SL2, all others had an *I_c0_*/*I_cprod_* value above 1. Except for lower *n-*values, no other difference could be observed between the initial properties of the “SD“ joints and the “SL“ joints.

#### 3.2.2. Thermal Cycling of the Joints

As mentioned in Section 2.2.3, we performed 135 TC with all the Group 2 joints without dismounting them from holders before each *I–V* measurement. We suppose that the latter modification (compared to Group 1) could reduce the thermal and mechanical degradation related to the minor TC.

Unlike in the case of Group 1, we prepared no cross-section joint samples parallel to the Group 2 joints for SEM examination. Instead, with all the Group 2 joints, we plan to perform a spatial monitoring of currents in the joints themselves, followed by the preparation of cross-sections from physically the same joints as used for *I–V* measurements, which should (unlike for Group 1) rule out the effects of variable SCT quality across the tape length. All the mentioned structural analysis of the Group 2 samples will be the subject of our next study.

##### 3.2.2.1. Evolution of R_j_/R_j0_

Analysis of the evolution of the normalized joint resistivity (*R_j_*/*R_j0_*) as a function of number of TC in Figure 9a shows that except for one measurement (the joint G2.SD1 after 135 TC), the *R_j_*/*R_j0_* did not rise by more than 37% of its initial value for any joint from Group 2. Among a total of 48 *I–V* measurements in Figure 9a, only four of them had *R_j._A_j_* values above the threshold of 50 nΩ·cm^2^; for half of the joints of Group 2, the *R_j._A_j_* never exceeded the 50 nΩ·cm^2^ limit. After all 135 TC, half of the joints even had *R_j._A_j_* values slightly lower than *R_j0._A_j_*. The median value of *R_j._A_j_* for all four joints (all measured points) was 7% below the *R_j0._A_j_*. The estimated change of *R_j._A_j_* per 1 TC by linear regression (each joint fitted separately in Appendix A) ranges from -0.04 to 0.47 (nΩ·cm^2^)/TC. The negative slope of linear fit was only observed for the joint G2.SL1, which seems to correlate with the improvement of its *I_c_* with the increasing number of TC. However, even in the case of raise of the *R_j_*/*R_j0_* value to the maximum of 2.99 (the joint G2.SD1 after 135 TC), it was not coupled with an unprecedented drop in the *I_c_*/*I_c0_*. The best performing joint, based on its lowest average of *R_j_*/*R_j0_* of 0.75, was the G2.SL2, for which (unlike the rest of the Group 2 joints) all normalized resistivities were lower than 1; thus, the primacy of the joint G2.SL2 would remain true even if its sharp minimum (which will be discussed separately) would be excluded from the averaging.

The effects of different amounts of minor thermal cycles coming from *I–V* measurements (15 minor TC for G2.SD1 and G2.SL1, versus nine minor TC for the joints with constant measuring step, that is G2.SD2 and G2.SL2) were negligible, as no clear difference could be observed between the two pairs of joints. The latter is further confirmed by the following measurement. After completing 135 TC, we wanted to verify the reproducibility of the *I–V* measurements themselves. The *I–V* dependence was measured eight more times for the joint G2.SL2 but with no LN_2_ → 150 °C TC in between the measurements (that is, only eight minor TC were performed, from room temperature to LN_2_), as displayed in all graphs of Figure 9 by red circles not connected by a line. (The joint G2.SL2 was selected for minor TC, as it had the highest standard deviation among the joints that showed no significant degradation after 135 TC.) Since the standard deviation of solely the eight mentioned minor TC for Figure 9a was 0.063, the outlined trends in the *R_j_*/*R_j0_* evolution as a function of number of TC may be comparable with the experimental noise, which should be analogically taken into consideration also for the evolution of *I_c_*/*I_c0_* and *n/n*_0_ discussed further.

Regarding the effect of flux, starting at 90 TC, the joints with the “SL“ flux showed slightly lower *R_j_*/*R_j0_* values compared to the “SD“ flux, which would be expected, since the “SL“ flux was designed specifically for lead-free solders. On the other hand, up to 60 TC, the joints with “SD“ flux seem to be more reproducible (the joint G2.SD1 compared with G2.SD2).

##### 3.2.2.2. Evolution of I_c_/I_c0_ and n/n_0_

The standard deviation value of 0.014 calculated for the eight measurements of *I–V* curves for the joint G2.SL2 without the LN_2_ → 150 °C thermal cycling in between (Figure 9b, red circles plotted outside of the chart area) suggests that the fluctuations mostly come from (or are the same order of magnitude as) the inherent experimental noise of the *I–V* measurements (analogically also for the evolution of *n/n*_0_ and *R_j_*/*R_j0_*). Therefore, we consider the trends in *I_c_*/*I_c0_* evolution to be rather flat, but we nevertheless try to estimate some trends in the *I_c_*/*I_c0_* evolution.

Figure 9b shows the dependence of the normalized critical current (*I_c_*/*I_c0_*) on the number of TC. There, 58% of measured points showed a slight degradation of *I_c_*. Nevertheless, during the whole course of 135 TC, for the joints G2.SD1–G2.SL2, we never observed a drop of *I_c_* below 85% of its initial value (*I_c0_*), and after completing all 135 TC, three out of four joints ended up with *I_c_*/*I_c0_* within an only +/- 2% deviation from their respective *I_c0_*. The median value of *I_c_* for all joints from Group 2 was 0.2% below the *I_c0_*. The joint G2.SD1 had by far the most scattered results of both the critical current and resistivity values. An increasing slope of linear fit to the *I_c_*/*I_c0_* versus TC dependence was only observed for the joint G2.SL1. The estimated change of *I_c_* per one TC by linear regression (each joint separately) ranges from -0.25 to +0.01 A/TC. The best performing joint, based on its highest average of *I_c_*/*I_c0_* of 1.015, was the G2.SL2. Overall, from the viewpoint of *I_c_*, all the Group 2 joints behaved reliably for the whole examined range of TC (Figure 9b), as all *I_c_* measurements were above the threshold for serious degradation (*I_c_* ≥ 0.8 *I_c0_*).

However, the above estimations of slopes could barely be used for the extrapolation of a joint lifetime, as the degradation will most likely manifest as an abrupt increase of *R_j._A_j_* and/or decrease of *I_c_*. If there are any early signs (at low number of TC) of the serious degradation of a joint’s *R_j._A_j_* later (at a high number of TC) on Figure 9, the scattering of the joint’s *I_c_* seems to forecast the increase of *R_j._A_j_* above 50 nΩ·cm^2^ better than changes of the evolution of *R_j._A_j_* itself (compare the joint G2.SD1 with G2.SD2–G2.SL2 in Figure 9b). On the contrary, the evolution of *R_j._A_j_* showed a similar TC evolution for differently behaving joints (regarding the serious degradation at highest amount of TC), comparing the joint G2.SD1 with e.g., G2.SD2 (Figure 9).

##### 3.2.2.3. Minimum of R_j._A_j_ at 30 TC in the Joint G2.SL2

In our previous thermal cycling study [24], after performing 27 TC, we found a decrease of the joint resistivity to only 44% of its initial value. To bring more insight into the occurrence of such a minimum, the frequency of *I–V* measurements for the joints G2.SD1 and G2.SL1 was increased in the range of 10 to 50 TC. Although only observed for one joint of four, at 30 TC, the *R_j._A_j_* of the joint G2.SL2 (paradoxically the one with a lower frequency of *I–V* measurements) indeed dropped dramatically from 49.4 nΩ·cm^2^ to 3.5 nΩ·cm^2^, which was only 7% of its value before the start of the TC, while the corresponding normalized critical current (*I_c_*/*I_c0_*) did not show any extreme value. Although the *I_c_* increased to 12% above its initial absolute value (*I_c0_*), which is the highest increase in Figure 9b, the *I_c0_* of the joint G2.SL2 was by far the lowest from all the Group 2 joints, and so the absolute *I_c_* value reached its maximum at only 92% of the average *I_c0_* of the remaining three joints (see the insert graph in Figure 9b). Moreover, from the Group 2 joints, the joint G2.SL2 had the highest *R_j0_*, the highest average of *I_c_*/*I_c0_* values, as well as the lowest coefficient of variance for *n/n_0_* values.

We ruled out the possibility that the decrease would be due to an experimental error by (1) double-checking the data evaluation, (2) comparing the *n*-value of this measurement, which was very close to the median for the other measurements of this joint, and (3) by comparison of our experimental value of 3.5 nΩ·cm^2^ with the (a) lowest found value for a similar joint in the literature, which was 3.6 nΩ·cm^2^ [10], and (b) a value of 1−2 nΩ·cm^2^, which is an estimation of resistivity for an ideal joint, which would have resistivity coming solely from the sum of the metallic layers themselves (no interface resistivity) [16]. It could be argued that we only recorded a significant minimum of *R_j._A_j_* for one out of four joints, and therefore, the reproducibility of the phenomenon is poor. However, the latter low probability of finding the minimum could result from making too big TC steps (the minima could be narrower than our measurements show).

Should there be a potential practical use for such a meta-stable (highly sensitive to TC) state of *R_j._A_j_*, in a following study, we could repeat the TC with even more fine cycling steps and relate the *R_j._A_j_* decrease to structural changes of the joint. In any case, we would draw a general recommendation for performing a measurement such as that shown on Figure 9 not to rely merely on one *I–V* measurement, as a very low value of *R_j._A_j_* could misrepresent the “normal“ joint TC behavior, especially in the vicinity of 30 TC.

All the measured *n-*values for the joint G2.SL2 were above 1 (Figure 9c); other *n-*values for Group 2 joints were decreasing in the course of TC, similarly to the Group 1 “SN” joints. On the other hand, the corresponding *I_c_/I_c0_* values (Figure 9b) were shown to be rather stable against the increasing number of TC.

### 3.3. Summary of AllJoints

Let us now summarize and compare the Group 1 and the Group 2 joints, and discuss some common features for all the investigated joints. No visible mechanical damage nor any unintended macroscopic de-soldering occurred during the whole study to any of the joints. The best achieved properties of *R_j._A_j_* among all the 15 joints were as follows: lowest *R_j0._A_j_* = 35.3 nΩ·cm^2^ (joint G2.SL1) and *R_j._A_j_* = 3.5 nΩ·cm^2^ (joint G2.SL2), in the initial state and during thermal cycling, respectively; joint G2.SL1 had the lowest *R_j._A_j_* = 38.5 nΩ·cm^2^ after 135 TC. A more detailed comparison can be inferred from Figure 10, in which the minima and maxima for a given number of TC were selected for the three curves, such that envelopes of data from Figure 4, Figure 8 and Figure 9 can be directly compared.

It could be argued that the “IN“ solder should have been used for the preparation of the Group 2 joints, in an attempt to further decrease the *R_j._A_j_*, based on a comparison of (1) the *R_j._A_j_* values of the Group 1 “SN“ and the Group 1 “IN“ joints in Figure 11 and (2) the electrical resistivities of the pure “IN“ solder and “SN“ solder in Table 3. However, there are two arguments against the latter hypothesis: (1) When we also considered the potential effect of the very high “IN“ solder porosity on the mechanical properties of the joints, which can (among other consequences) negatively affect the number of TC before serious degradation, we assumed the “SN“ solder as a better option overall. (2) As mentioned above, the contribution of a solder interface resistivity might be an order of magnitude higher than the resistivity of the solder layer itself, especially for the “IN“ solder, as it does not have an ideal solderability with copper surfaces.

Based on the overview of our results, we propose several ways to improve the performance of the soldered joints in future research: (1) To increase the accuracy of measurements presented in Figure 4, Figure 8 and Figure 9, the first measure would be to reduce the experimental noise of the *I–V* measurement itself and from fitting of the raw *I–V* curves, such that the fluctuation contributions coming from (a) the weak structural changes in the soldered layer and (b) random experimental errors would dominate. In the second step, we would attempt to distinguish between the two latter mentioned fluctuation components. (2) For the selection of a suitable SCT, a non-destructive technique could be employed for the microstructural analysis of internal interfaces of the SCT part to be overlapped. (3) After a joint fabrication, its microstructure could be correlated with a spatial distribution of the joint resistivity, instead of the measurement of only one average value of *R_j_.Aj.* (4) While the rate of heating used throughout our study was in the order of 10^0^ °C/s (Table 6), it might exceed 10^5^ °C/s for some applications, e.g., an SFCL. A more appropriate heating rate of joints during thermal cycling could be ensured, e.g., by using a heat-transfer medium with high thermal capacity instead of air. Under the latter conditions, the joint would experience a much shorter period during which the soldered layer dwells above the room temperature, which could support the unwanted β-form metallic (Sn) transformation to the non-metallic α-(Sn), and thus, it could cause an increase of the joint resistivity (5) Our *I–V* measurements were performed in self field only. Insight into the effects of the magnetic field applied to the tapes in longitudinal and perpendicular directions could broaden the range of joint applications. (6) Alternative approaches for the preparation of the solder layer could be tested, reducing the joint interface resistivities, e.g., in situ magnetron sputtering of solder components (Sn, Cu, Ag) directly onto the (RE)BCO surface.

## 4. Conclusions

By the investigation of two groups of soldered joints, we showed that it is possible to effectively prepare joints with resistivities reliably below 50 nΩ·cm^2^, with good resilience against tens of thermal cycles spanning hundreds of °C. However, great attention must be paid to the quality of interlayer interfaces of the overlapped parts of the SCT. While in the Group 1 joints, the In-based solder showed relatively lower resistivities, the Sn-based solder proved to be more appropriate for the soldering of Cu-finished tapes. Nevertheless, due to suspected high SCT interface resistivities, almost all the joint resistivities in Group 1 were in the order of 10^2^ nΩ·cm^2^, showing only little dependence on the used materials and/or preparation conditions. On the other hand, in case solely SCT with no apparent interface structural defects were used (the Group 2 joints), three out of four joints showed resistivities in the order of 10^1^ nΩ·cm^2^, even after 135 thermal cycles, with no significant degradation of the critical current.

## Figures and Tables

**Figure 1 materials-14-01052-f001:**
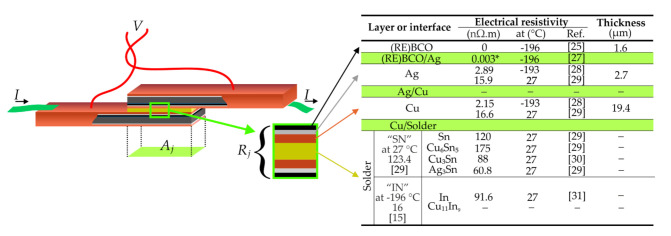
Scheme of a joint with overlap area *A_j_* and overall (layer and interface) joint resistance *R_j_*. Layer thicknesses are not in scale. Composition and properties of the “IN” and “SN” solders will be listed further (Table 3). Current*–*voltage measurements were performed with the four-probe method, where dark green wires are current terminals (*I*), and red wires are voltage contacts (*V*). Inserted table with layer properties: light green rows are denoting layer interfaces; * = units nΩ·m^2^. Where resistivities at boiling temperature of nitrogen were not found in the literature, values at higher temperatures are listed.

**Figure 2 materials-14-01052-f002:**
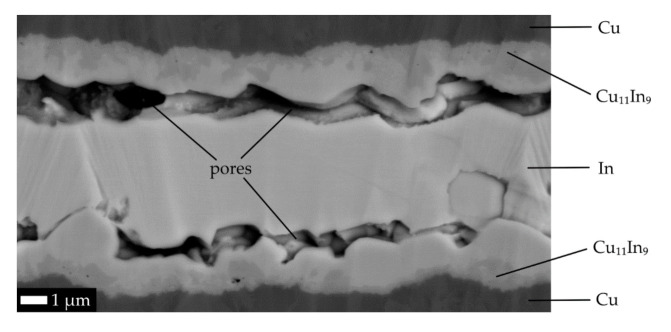
SEM micrograph of soldered joint G1.IN3 in cross-section. Chemical composition of layers was determined by energy-dispersive X-ray (EDX) map and/or point spectra.

**Figure 3 materials-14-01052-f003:**
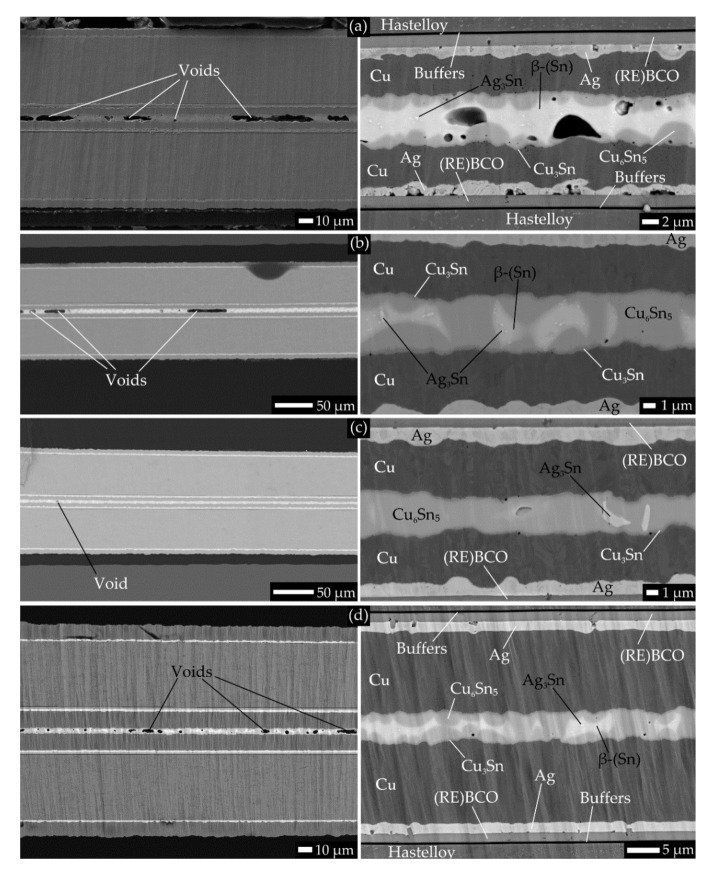
Cross-section SEM micrographs with a typical structure of the soldered joints using the “SN“ solder (Group 1). On the left hand side, an overview, and on the right hand side, a magnified image of cross-sectioned samples: (**a**) The joint G1.SN1 with a visible uncompact Ag layer and voids in the joint; (**b**) the joint G1.SN5 and (**c**) the joint G1.SN6 with a dominant Cu_6_Sn_5_ phase in a thin joint. The joint G1.S6 with improved joint resistivity (**d**). The chemical composition of layers was determined by EDX map and/or point spectra.

**Figure 4 materials-14-01052-f004:**
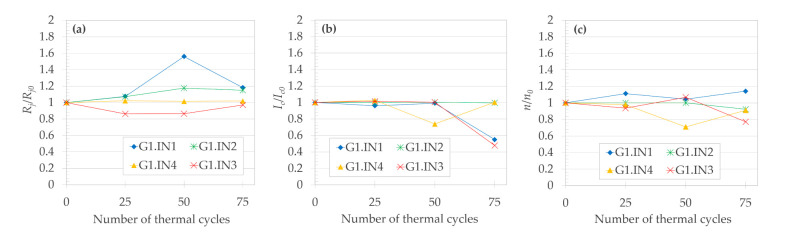
Normalized values of (**a**) joint resistivity, (**b**) critical current, and (**c**) *n*-value of the Group 1 joints soldered with the “IN“ solder, as a function of number of thermal cycles.

**Figure 5 materials-14-01052-f005:**
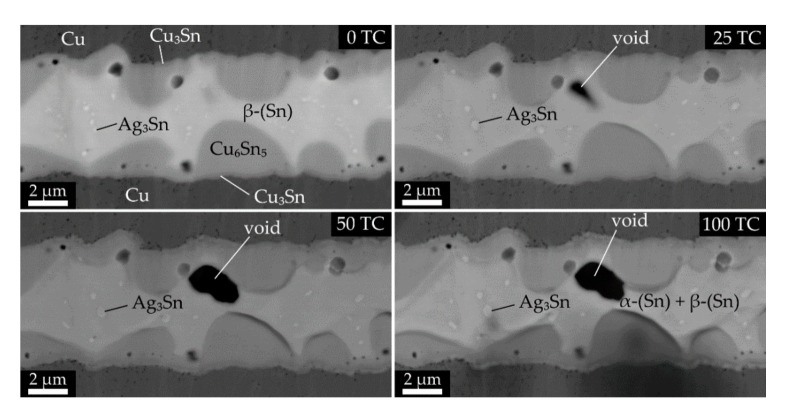
Cross-section SEM micrographs of the G1.SN1 joint with changes in microstructure such as void formation, enlargement of Ag_3_Sn particles, and phase transition of β-Sn to α-Sn, which was manifested by swelling within the solder layer developed during the thermal cycling. The chemical composition of layers was determined by EDX map and/or point spectra.

**Figure 6 materials-14-01052-f006:**
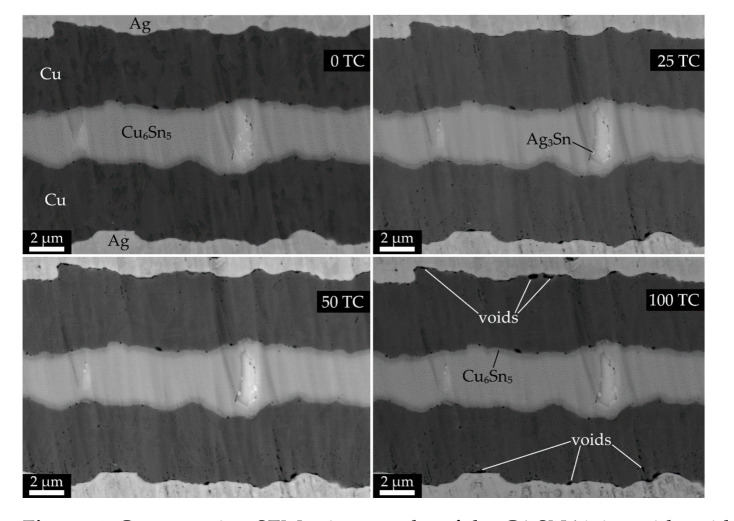
Cross-section SEM micrographs of the G1.SN6 joint with void formation at the Ag/Cu interface during thermal cycling. The chemical composition of layers was determined by EDX map and/or point spectra.

**Figure 7 materials-14-01052-f007:**
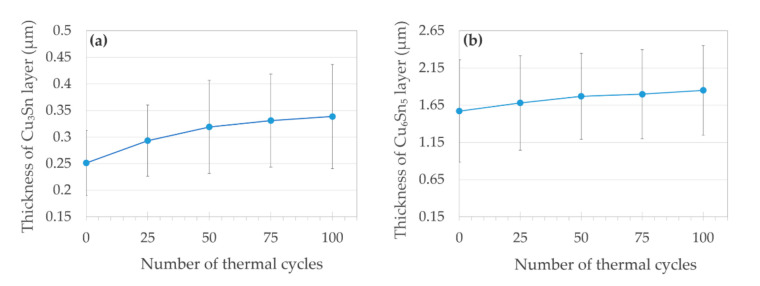
Thickening of intermetallic layers during the thermal cycling in the soldered joint G1.SN6: (**a**) Cu_3_Sn and (**b**) Cu_6_Sn_5_.

**Figure 8 materials-14-01052-f008:**
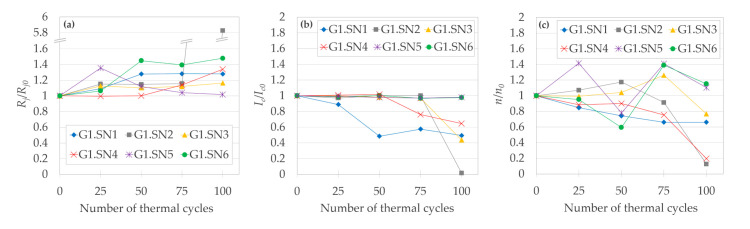
Normalized values of (**a**) joint resistivity, (**b**) critical current, and (**c**) *n*-value of Table 1. joints soldered with the “SN“ solder, as a function of number of thermal cycles.

**Figure 9 materials-14-01052-f009:**
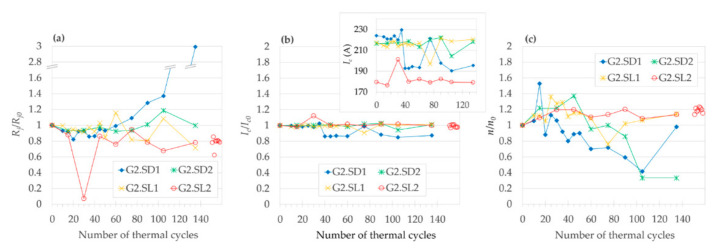
Normalized values of (**a**) joint resistivity, (**b**) critical current, and (**c**) *n*-value of the Group 2 good-quality joints, as a function of number of thermal cycles. Red circles not connected by a line (at the rightmost part of plot) are minor TC of the joint G2.SL2; the horizontal scale does not apply for these points.

**Figure 10 materials-14-01052-f010:**
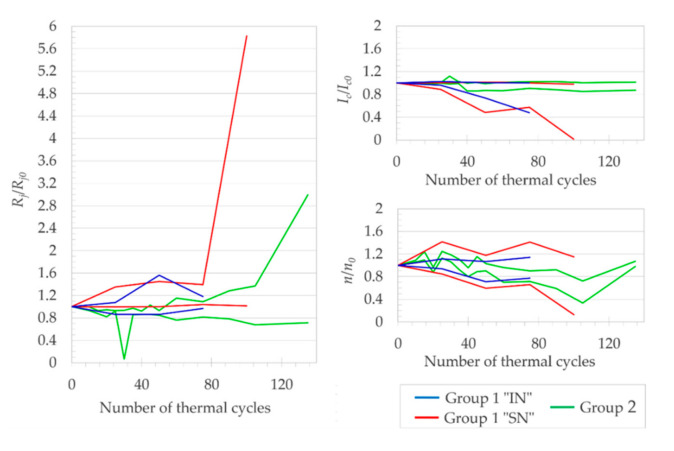
Comparison of minimal and maximal values of Group 1 (Figure 4 and Figure 8) and Group 2 (Figure 9) joints.

**Figure 11 materials-14-01052-f011:**
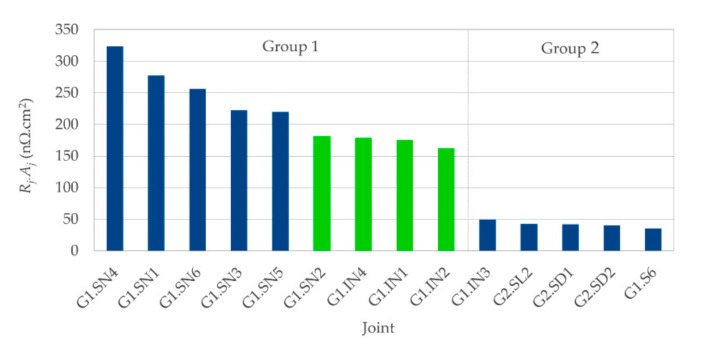
Resistivities of all joints (Group 1 and Group 2), sorted in descending order. Blue columns are joints with the “SN“ solder, green columns are for the “IN“ solder.

**Table 1 materials-14-01052-t001:** Literature overview of joint resistivities. Only initial states of (RE)BCO Cu-finished superconducting tapes (SCT) soldered joints were selected. In case multiple values were found, range (minimum–maximum) is reported. For better readability, resistivities were rounded to integers. Sorted according to increase of the lower limit (in bold) of the range. Different SCT producers listed in columns (SHSC = Shanghai Superconductor). *R_j_* = joint resistance, *A_j_ =* joint overlap area (see also Figure 1).

Composition of Used Solder	Joint Resistivity *R_j._A_j_* (nΩ·cm^2^)						Ref.
	SuperPower	SuNAM	SuperOx	AMSC	SHSC	Self-Made	
InBi	4	–	–	–	–	8	[10]
Sn63Pb37	5–36	–	–	–	–	–	[11]
Sn61Pb39	12–33	45–88	8–13	–	–	–	[12]
In80Pb15Ag5	9–648	–	–	–	–	–	[13]
SnIn, SnPb	17–671	–	–	73–880	–	–	[14]
In66Bi34, In52Sn48,In97, Sn63Pb37	–	–	–	–	–	20–66	[15]
In52Sn48, Pb63Sn37	–	–	–	21–880	–	–	[1]
Sn60Pb40	60–77	124–137	27–49	–	26–41	–	[16]
In alloy, Sn60Pb40In80Pb15Ag5	28–1093	–	–	–	–	–	[17]
SnBi	28–52	–	–	–	–	–	[18]
In100, In52Sn48, Bi67In33	–	–	–	29–684	–	–	[19]
Pb38Sn62, In52Sn48	30–50	–	–	–	–	–	[20]
Sn63Pb37	37–1746	–	–	–	–	–	[21]
Sn96.5Ag3Cu0.5, Sn99Ag0.3Cu0.7	41–81	–	–	–	–	–	[22]
Sn52In48, Sn61Pb39	60–133	–	168–174	–	–	–	[23]
Sn96.5Ag3Cu0.5	67–164	–	–	–	–	–	[24]
InBi	–	–	–	300–7047	–	–	[25]

**Table 2 materials-14-01052-t002:** Properties of superconducting tapes used for preparation of the soldered joints. *I_c_* = critical current. The general definition of the *n*-value can be found e.g., in [24] or [36]. Indices “prod” denote values provided by the tape producer. Thicknesses of the Cu and Ag overlayers denote the “top” layers, that is those closer to the (RE)BCO layer; see also Figure 1.

Abbrev.	Type	*I_cprod_*(A)	*n_prod_*	Width (mm)	Layer Thickness (µm)				
					Cu (top)	Ag (top)	(RE)BCO	Buffer	Hastelloy
S4	SCS4050-AP	117	30	4.01	9.67 ± 0.17	1.24 ± 0.07	1.43 ± 0.01	0.26 ± 0.01	46.38 ± 0.14
S6	SCS6050-AP	215	29	6.05	12.43 ± 0.24	1.50 ± 0.16	1.40 ± 0.03	0.29 ± 0.01	45.45 ± 0.13

**Table 3 materials-14-01052-t003:** Properties of solders used for preparation of the tape–tape joints (IN, SN) and for the tape–terminals (FL) joints: *R_s_ =* electrical resistivity, *T_liquidus_* = liquidus temperature of the solder. Except for the values of solder resistivities (see also Figure 1), all data were provided by the respective solder producer.

Abbrev.	Commercial Name	Chem. Composition	*R_s_* (nΩ.m)	*T_liquidus_* (°C)	Supplier	Note
IN	–	In97Ag3	75 at 20 °C [39]	143	“Research Kit” by Indium Corp.	Low resistivity
SN	SAC305	Sn96.5Ag3Cu0.5	123.4 at 27 °C [29]	220		Good solderability with Cu
FL	Fluitin 1532	Sn60Pb38Cu2	–	183	Alpha Metals	Only for SCT to tape terminals joints

**Table 4 materials-14-01052-t004:** Properties of fluxes. *T_activation_* = flux activation temperature.

Abbrev.	Commercial Name	Type	*T*_activation_ (°C)	Consistence	Supplier	Deposition
SD	SMD291	Synthetic, no-clean	140	Thick paste	Chip Quik, NY, USA	Spread onto by plastic tip
SL	SMD291NL	Synthetic, no-clean	140	Thick paste	Chip Quik, NY, USA	Spread onto by plastic tip
F2	FLUX-2	–	100–371	Liquid	Indium Corp.	As a thin film by a brush
RN	–	High-purity rosin	–	Solid	AV-EL mak., Slovakia	As finely ground powder, sintered at 80 °C for 10 min
R5	–	50 wt% of “RN” dissolved in 99.8 vol% ethanol	–	Liquid	Self-made	By pipette, solvent evaporated at room temperature

**Table 5 materials-14-01052-t005:** Overview of the prepared joints. *T_q_*
_=_ quenching temperature, *t_l→q_* = reflow time.

Jointlabel	Joint Materials			Cleaning	Soldering Conditions		
	SCT	Solder	Flux		Pressure (kPa)	Tq (°C)	tl→q (s)
G1.IN1	S4	IN	F2	Nital	309	160	45
G1.IN2	S4	IN	F2	Nital	309	160	45
G1.IN3	S4	IN	SD	Nital	309	180	77
G1.IN4	S4	IN	F2	Acetone	397	180	45
G1.SN1	S4	SN	F2	Nital	397	240	38
G1.SN2	S4	SN	R5	Nital	397	240	39
G1.SN3	S4	SN	F2	Acetone	442	245	40
G1.SN4	S4	SN	RN	Nital	175 → 309 *	219 → 240 *	100 → 170 *
G1.SN5	S4	SN	RN	Nital	175 → 309 *	220 → 240 *	50 → 60 *
G1.SN6	S4	SN	SL	Nital	309	245	87
G1.S6	S6	SN	SL	Nital	292	245	70
G2.SD1	S6	SN	SD	Nital	292	245	80
G2.SD2	S6	SN	SD	Nital	292	245	65
G2.SL1	S6	SN	SL	Nital	292	245	–
G2.SL2	S6	SN	SL	Nital	292	245	66

* = Two step reflow, see details in Section 3.1.1.2.

**Table 6 materials-14-01052-t006:** Timings and temperatures experienced by the joints during their thermal cycling.

TC Stage		Heating in Oven	Immersion in LN_2_	At room Temperature
Temperature (°C)		≤ 150	150 → −196	≥ −196
Dwell time (s)	Group 1	60	10	10
	Group 2	600	60	60

**Table 7 materials-14-01052-t007:** Overview of initial measurements for Group 1 of the soldered joints. The indices “0” denote the state at 0 TC, that is, before thermal cycling.

Joint Label	*I–V* Measurements				Soldered Layer	
	*R_j0._A_j_* (nΩ.cm^2^)	*I_c0_*(A)	*I_c0_/I_cprod_* (%)	*n_0_*	Thickness (µm)	Porosity (vol%)
G1.IN1	178.9	109.4	94	24	12.8	85
G1.IN2	175.3	103.9	89	22	12.8	85
G1.IN3	162.5	99.6	85	22	9.5	90
G1.IN4	181.3	120.9	103	32	11.0	60
G1.SN1	323.4	123.2	105	31	6.5	75
G1.SN2	220.1	115.2	98	23	8.1	95
G1.SN3	256.3	121.4	104	29	6.4	85
G1.SN4	340.7	117.5	100	31	8.5	20
G1.SN5	222.4	116.6	100	26	5.5	20
G1.SN6	277.6	93.3	80	21	3.3	10
G1.S6	40.6	221.3	103	31	3.0	25

**Table 8 materials-14-01052-t008:** Overview of initial measurements for the Group 2 soldered joints. The indices “0” denote the state at 0 TC, that is, before thermal cycling.

Joint Label	*I–V* Measurements			
	*R_j0._A_j_* (nΩ.cm^2^)	*I_c0_* (A)	*I_c0_/I_cprod_* (%)	*n_0_*
G2.SD1	42.9	224.0	104	25
G2.SD2	42.2	216.5	104	25
G2.SL1	35.3	217.5	101	20
G2.SL2	49.4	179.7	84	18

## Data Availability

The data presented in this study are available on request from the corresponding author.

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
