# Peer review of "Thermal Cycling of (RE)BCO-Based Superconducting Tapes Joined by Lead-Free Solders"

_materials, 2021, doi:10.3390/ma14041052_

Round 1

Reviewer 1 Report

The authors presented the experimental work focusing on joints for superconducting wire connections. They systematically investigated how the fabrication methods influence the performance of the joints, and then evaluated the performance after thermal cycles. The study separated samples into two groups: group (1) was prepared using various solder, flux, cleaning agent, and materials; group (2) is the high-performance group as the best one from the group (1). The authors observed that the SCT interface resistivity has the largest influence on the performance of the soldered joints and the rate of critical current degradation. This manuscript is well written and clearly structured. I would recommend publication after minor revisions on the following points:

-Could the authors discuss more the fluctuations showing in the thermal cycle plots? The authors have discussed the overall trends, but what are the causes for the fluctuation? Should the authors take more intermediate measurements to ensure accuracy? Please justify.

-There are other minor issues:

(1)The authors should combine some of the paragraphs in the introduction section. It is hard to read when one sentence is made into a paragraph frequently.

(2)"...used to connect SCT a with (1) conventional conductors..." Please remove the "a" before with.

(3)The authors should explain each parameter presented in each table in the table caption.

(4)The authors should construct supplemental information to include the "not shown" plots.

Author Response

Point 0: Could the authors discuss more the fluctuations showing in the thermal cycle plots? The authors have discussed the overall trends, but what are the causes for the fluctuation? Should the authors take more intermediate measurements to ensure accuracy? Please justify.

Response 0: We added discussion about the fluctuations to section 3.2.2.2. We also added a new point (number 1) discussing the causes for the fluctuations, as well as the intermediate measurements to ensure accuracy, to the last paragraph of section 3.3.

Point 1: The authors should combine some of the paragraphs in the introduction section. It is hard to read when one sentence is made into a paragraph frequently.

Response 1: We reduced the number of paragraphs in the "Introduction" section from the original 22 to the current 11 by groupping similar references to one paragraph.

Point 2: "...used to connect SCT a with (1) conventional conductors..." Please remove the "a" before with.

Response 2: Revised as requested.

Point 3: The authors should explain each parameter presented in each table in the table caption.

Response 3: We added explanations to the captions of Tables 1, 2 and 5.

Point 4: The authors should construct supplemental information to include the "not shown" plots.

Response 4: Please see the attached copy of the supplementary material - we included eight plots there (Figures S3-S10), which were "not shown" in the article. The "not shown" phrases in the article were replaced with the respective references to the supplemental figures.

Reviewer 2 Report

Authors will need to address the following comments in the manuscript:

  1. In Introduction, please generally explain some applications in superconductivity which need joints.
  2. Please add some picture of your experimental set up, as well as your sample photos, if any.
  3. Why did you choose this type of thermal cycling? Is there any standard way of doing it?
  4. Will heating up speed affect the quality of joint drastically?

Author Response

Point 1: In Introduction, please generally explain some applications in superconductivity which need joints.
Response 1: We added 6 applications to the second paragraph of the "Introduction" section.

Point 2: Please add some picture of your experimental set up, as well as your sample photos, if any.
Response 2: Please see the attached copy of the supplementary material. We added a description of our experimental setup (Figure S1) and some examples of soldered joints (Figure S2). Both added figures were also appropriately referenced in the text.

Point 3: (a) Why did you choose this type of thermal cycling? (b) Is there any standard way of doing it?
Response 3:
(a) The type of thermal cycling we used was a compromise between two requirements: (1) To ensure conditions close to the ones expected for the most probable applications of joints, that is in a superconducting fault current limiter, and (2) to use a method potentially usable in industry, with stress to the time requirements and costs. We included this response into the section 2.2.3.
(b) We are not aware of any standard for performance of thermal cycling of superconducting tapes; we hope that conclusions of our publication will be useful should a standard will be developped.

Point 4: Will heating up speed affect the quality of joint drastically?
Response 4: The effect of the increased heating rate would indeed be an important issue to investigate in our future work. Although the producers of superconducting tapes try to ensure that all of the layers of the tape sandwich structure would have similar coefficients of thermal expansions, in case 100 000 times faster temperature increase is used (compared to our conditions), undesired structural changes can be forecasted. After tens to hundreds of thermal cycles, the aging of tapes themselves as well the soldered layer could possibly lead to increase of the joint resistivity, and decrease of the joint mechanical properties at the same time. The latter deterioation could be potentially enhanced by much shorter period during which the soldered layer dwells above the room temperature, which would probably support the unwanted β-form metallic (Sn) transformation to the non-metallic α-(Sn). We modified the section 3.3 according to these assumptions.

Reviewer 3 Report

referee report 
materials-1115993-peer-review-v1
Thermal Cycling of (RE)BCO-based Superconducting Tapes Joined by Lead-Free Solders
Michal Skarba1, Marcela PekarÄŤíková, Lubomír Frolek, Eva Cuninková and Martin Necpal 

This manucript reports on the fabrication and characterization of joints between superconducting YBCO tapes using
lead-free solders. This topic is very important for several applications of superconducting tapes, where joints
between the tape pieces are unavoidable. Thus, the outcome of this research plays a key role in this field.
The present manuscript is well written and well organized, both scientifically and technically. The manuscript
comprises 11 figures, 8 tables and gives 51 references. The references provided give a full overview of the 
activities of the community in the last years.
The introduction of the manuscript is well prepared, giving a good overview of the topic including a table with
valuable data collected by the authors. All figures are well prepared.
All the steps carried out by the authors are properly explained and discussed.
The English requires only slight improvement (use of defined and undefined articles).
In the reference list, the correct abbreviation of IEEE TAS is "IEEE Trans. Appl. Supercond."

Of course, it will be interesting to see the effects of magnetic field applied to the tapes in longitudinal and
perpendicular directions. This will certainly be a topic for a future paper.

Thus, this manuscript may be published with minor revision.

Author Response

Point 1: The English requires only slight improvement (use of defined and undefined articles).
Response 1: Revised as requested, we added multiple missing "a/an" and "the" articles.

Point 2: In the reference list, the correct abbreviation of IEEE TAS is "IEEE Trans. Appl. Supercond."
Response 2: All 18 occurences were revised as requested.

Point 3: Of course, it will be interesting to see the effects of magnetic field applied to the tapes in longitudinal and perpendicular directions. This will certainly be a topic for a future paper. 
Response 3: The proposed measurement could indeed significantly broaden the range of joint applications, we therefore added this idea to the list of future work - point (5) in the last paragraph of section 3.3.